# Improved Atrial Differentiation of Human Pluripotent Stem Cells by Activation of Retinoic Acid Receptor Alpha (RARα)

**DOI:** 10.3390/jpm12040628

**Published:** 2022-04-13

**Authors:** Verena Schwach, Carla Cofiño-Fabres, Simone A. ten Den, Robert Passier

**Affiliations:** 1Department of Applied Stem Cell Technologies, TechMed Centre, University of Twente, 7522 NB Enschede, The Netherlands; c.cofinofabres@utwente.nl (C.C.-F.); s.a.tenden@utwente.nl (S.A.t.D.); 2Department of Anatomy and Embryology, Leiden University Medical Center, 2300 RC Leiden, The Netherlands

**Keywords:** atrial cardiomyocyte differentiation, hPSC-derived AM, retinoic acid signaling

## Abstract

Human pluripotent stem cell (hPSC)-derived cardiomyocytes have proven valuable for modeling disease and as a drug screening platform. Here, we depict an optimized protocol for the directed differentiation of hPSCs toward cardiomyocytes with an atrial identity by modulating the retinoic acid signaling cascade in spin embryoid bodies. The crucial steps of the protocol, including hPSC maintenance, embryoid body (EB) differentiation, the induction of cardiac mesoderm, direction toward the atrial phenotype, as well as molecular and functional characterization of the cardiomyocytes, are described. Atrial cardiomyocytes (AMs) can be generated within 14 days. Most importantly, we show that induction of the specific retinoic acid receptor alpha (RARα) increased the efficiency of atrial differentiation to 72% compared with 45% after modulating the retinoic acid (RA) pathway with all-trans RA (atRA). In contrast, the induction of RARβ signaling only had a minor impact on the efficiency of atrial differentiation (from about 45% to 50%). Similarly, the total yield of AM per EB of 5000 hPSCs was increased from 10,350 (2.07 per hPSC) to 16,120 (3.22 per hPSC) while selectively modulating RARα signaling. For further purification of the AMs, we describe a metabolic selection procedure that enhanced the AM percentage to more than 90% without compromising the AM yield (15,542 per EB, equal to 3.11 per hPSC) or functionality of the AMs as evaluated by RNAseq, immunostaining, and optical action potential measurement. Cardiomyocytes with distinct atrial and ventricular properties can be applied for selective pharmacology, such as the development of novel atrial-specific anti-arrhythmic agents, and disease modeling, including atrial fibrillation, which is the most common heart rhythm disorder. Moreover, fully characterized and defined cardiac subtype populations are of the utmost importance for potential cell-based therapeutic approaches.

## 1. Introduction

Cardiac diseases are a major societal burden with a high impact on mortality, morbidity, as well as financial costs [1]. Clearly, realization of successful therapeutics is facilitated by a better understanding of the underlying disease mechanisms, as well as by suitable platforms for drug development and screening. Although expenditure on drug discovery programs has increased significantly over the last decade, the number of new medicines that became available for treatment of patients is dissatisfying. Lack of predictivity of currently used animal and non-human in vitro models, combined with the difficulty to obtain and maintain human primary cultures like human cardiomyocytes (CMs), are major reasons for this inefficient process. This problematic situation has instigated research using human-induced pluripotent stem cells (control or patient-derived) for disease modeling, drug discovery, and regenerative medicine. Human pluripotent stem cells (hPSCs) have the capability to differentiate into all cells of the human body, including various CM subtypes of the heart. HPSC-derived CMs with ventricular identities (VMs) have successfully been applied for disease modeling and drug testing during the last decade [2,3]. Only recently has differentiation to human atrial CMs (AMs) from hPSCs been described [4,5,6,7]. Using CRISPR/Cas9, we previously generated an NKX2.5^eGFP/+^-COUP-TFII^mCherry/+^ fluorescent hPSC reporter to identify and select AMs as well as VMs by inserting the sequences of red fluorescent mCherry into the genomic locus of chicken ovalbumin upstream promoter transcription factor II (*COUP-TFII)*, an atrial enriched transcription factor, within the well-established *NKX2.5*^eGFP^ hPSC line. With this line, AMs can be selected by NKX2.5-GFP (G^+^) and COUP-TFII-mCherry (M^+^) expression, whereas VMs can be purified by NKX2.5-GFP (G^+^) expression and the absence of COUP-TFII-mCherry (M^−^) expression (Figure 1a) [8]. Defined differentiation and purification of AMs and VMs are very important for drug discovery and validation, since cardiac subtypes may differently affect cardiac disease or respond to drugs. For example, current anti-arrhythmic compounds for the treatment of atrial fibrillation, the most common arrhythmia, are not specific for the atrium and risk fatal cardiac effects on the ventricles [9]. Thus, there is an urgent need for selective pharmacology utilizing human CMs with atrial identity. Similarly, for regenerative purposes, it is crucial to have robust, standardized, and efficient protocols for the production of human AMs and VMs.

### Overview of Protocols to Generate hPSC-Derived AMs

We have previously described the generation of AMs and VMs by modulating the retinoic acid (RA) signaling cascade during differentiation of hPSCs (Figure 1b) [7,8]. The expression of atrial-specific genes, including the ion channel genes *KCNA5* (encoding K_v_1.5 channels) and *KCNJ3* (encoding Kir 3.1) was significantly increased in hPSC-AMs when compared with their ventricular counterparts. In addition, we showed by short hairpin RNA (shRNA)-mediated knockdown and chromatin immunoprecipitation that these ion channel genes were partially regulated by atrial-enriched COUP-TF transcription factors I and II [7]. In corroboration, whole-genome microarray analysis of the hPSC-derived AMs and VMs were compared to human fetal and adult atrial and ventricular samples, which confirmed their identities [7,10]. Moreover, analysis by patch clamp electrophysiology of hPSC-derived AMs in the presence of atrial-specific compounds (carbachol and 4-aminopyridine) displayed, as expected, functional responses in the AMs, whereas the VMs were unresponsive [7]. In recent years, several protocols for the derivation of AMs by modulating RA signaling have been developed [4,5,6,7,11]. Based on previous work with mouse ESCs [11], Zhang et al., showed that 1 µM of RA treatment in combination with BMP inactivation during differentiation of hPSCs resulted in high percentages of CMs with an atrial phenotype [6]. By alternating RA signaling at the mesodermal stage of hPSC differentiation, it was shown that AMs and VMs develop from distinct cardiac mesoderm progenitors [5]. Pei et al. developed a chemical-defined and albumin-free differentiation protocol using small molecules for the formation of cardiac mesoderm, followed by RA treatment from day 5 to 8 to induce AM differentiation [4].

Here, we describe an optimized and robust step-by-step spin embryoid body (EB) protocol to direct differentiation of hPSCs towards AMs by selectively inducing RARα. We show that the administration of the RARα selective agonist BMS-753 improved the atrial differentiation of hPSCs when compared to RA (Figure 1c, Appendix A). For the generation of pure populations of AMs, we introduced metabolic selection of RARα-treated cultures using lactate as the major nutrition factor in the medium. We describe crucial steps, expected outcomes, and troubleshooting. We deeply characterized the generated AMs by RNAseq, immunostainings and optical membrane potential recording.

## 2. Materials and Methods

### 2.1. Culture and Mantainance of hPSCs

HES3 COUP-red (NKX2.5^eGFP/+^-COUP-TFII^mCherry/+^) hESC and LUMC0020iCTRL-06 hiPSC cells were maintained as undifferentiated colonies on feeder-free defined cultures with Essential 8 (E8) stem cell medium (Appendix A) and vitronectin as a coating reagent. The hPSCs in E8 were passaged with EDTA twice a week (Figure 2c) and were cryopreserved in E8 cryopreservation medium (Appendix A). Alternatively, hPSCs were maintained as undifferentiated colonies on irradiated mouse embryonic feeder (MEF) cells in hPSC medium and were enzymatically passaged with 1x TrypLE Select twice a week (Appendix A). See Appendix A for detailed materials and a step-by-step procedure.

### 2.2. Generation of EBs from hPSCs

HPSCs grown in E8 can be differentiated toward AMs via spin EB differentiation by resuspending them in the EB formation medium (Appendix A) and aggregating them to spin EBs one day before the induction of cardiac mesoderm (Figure 1b). For EB formation from the hPSCs on the MEFs, the hPSCs were seeded on Matrigel one day before the start of differentiation for MEF cell depletion. The following day, the hPSCs were dissociated and resuspended in cardiac mesoderm induction medium before aggregation by centrifugation, as described previously (Figure 1b) [7,8]. See Appendix A for detailed materials and a step-by-step procedure.

### 2.3. Induction of Cardiac Mesoderm by Growth Factor-Induced Differentiation of hPSCs

According to previously published protocols, cardiac mesoderm was induced in the presence of a cocktail of growth factors composed of activin-A, bone morphogenetic protein 4 (BMP4), the small molecule inhibitor of glycogen synthase kinase-3β (CHIR-99021), vascular endothelial growth factor (VEGF) and stem cell factor (SCF) in BPEL medium for the initial 3 days of differentiation [6,7,12,13] (Appendix A, Figure 1, and Appendix A).

### 2.4. Induction of an Atrial or Ventricular Phenotype

On day 4 of differentiation, the EBs were refreshed with either all-trans RA (atRA) or the RARα selective agonist BMS-753 (Appendix A). For induction with RARα, the EBs were refreshed after 24 h with plain BPEL. At day 7 of differentiation, the EBs from both induction methods were plated on 0.1% gelatin-coated wells in BPEL medium. Optionally, the EBs could be refreshed with BPEL without plating. The EBs were refreshed again with BPEL at day 10 (Figure 1, Appendix A). The EBs could either be kept in BPEL until further characterization between day 14 and 21 or metabolically purified as described in Section 2.5. Alternatively, for the generation of ventricular CMs (VMs), the EBs were kept in plain BPEL from day 3 until day 7 (continuing as indicated in Appendix A).

### 2.5. Enrichment of AMs Based on Metabolic Selection

The beating AMs at day 13 were dissociated with 10X TripLE Select and plated on vitronectin-coated well plates with CM medium supplemented with 3′,5-Triiodo-L-thyronine sodium salt (T3, T), Dexamethasone (D) and Long R3 IGF-I (I) (TDI) (Appendix A) prior to lactate purification. The next day, metabolic selection was started by switching the medium to a glucose-free CM medium plus TDI supplemented with sodium DL-lactate (Appendix A). After 3 days, selection was stopped by refreshing the purified AMs with cardiomyocyte medium plus TDI. See Appendix A for the materials and a detailed step-by-step procedure.

### 2.6. Characterization of AMs

#### 2.6.1. Flow Cytometry Analysis or FACS

The dissociated and filtered samples were immediately analyzed for determining the percentage of GFP- and mCherry-positive AMs with a flow cytometer or for sorting AMs by FACS. Subsequent data analysis was performed with FlowLogic software analysis, and the results were expressed as the percentage of positive or negative cells. See Appendix A for the materials and a detailed step-by-step procedure.

#### 2.6.2. Immunotyping and Confocal Imaging

The dissociated samples or purified AMs were seeded on vitronectin-coated glass bottom plates or coverslips and fixed with 2% paraformaldehyde for 20 min at room temperature (RT), followed by permeabilization with 0.1% Triton X100 for 8 min at RT. After blocking for 1 h at RT with 5% fetal bovine serum (FBS), the cells were incubated at 4 °C overnight with the primary antibodies COUP-TFI, COUP-TFII, cTnI, or NKX2.5, as indicated in Appendix A. After washing, the cells were incubated for 1 h at RT with secondary antibodies AF488 or Cy3 and 5 min at RT with DAPI, and the coverslips were mounted with ProLong-Gold. Confocal images were captured using a Leica SP5 or Nikon Eclipse A1. See Appendix A for more information.

#### 2.6.3. Optical Membrane Potential Imaging

For optical membrane potential imaging, the sorted AMs or VMs at day 14 were plated on vitronectin-coated glass bottom plates in CM+TDI medium. At day 21, the cells were stained for 30 min at RT with the voltage-sensitive dye FluoVolt™ according to the manufacturer’s procedure. The excitation and emission wavelength of the dye were 522 and 535 nm, respectively. For more information about sample preparation, see Appendix A. For signal acquisition, the loaded sample was excited using a 488-nm laser and recorded for 10 s. The movies were filtered, and thereafter, time sequence plots of the fluorescence intensity over time were generated and used for calculation of the frequency and action potential duration at 20, 50, and 90% repolarization (APD_20_, APD_50_, and APD_90_, respectively) with a custom-written R script.

#### 2.6.4. Quantitative Real-Time PCR (RT-qPCR)

The RNA from the AMs or VMs was purified using a NucleoSpin RNA kit (Macherey-Nagel, Düren, Germany) according to the manufacturer’s protocol and reverse transcribed to cDNA using an iScript cDNA Synthesis kit (Bio-Rad, Lunteren, The Netherlands). Gene expression was assessed using a Bio-Rad CFX384 real-time system with SensiMix SYBR (Meridian Bioscience, Boxtel, The Netherlands). The samples were analyzed with Bio-Rad CFX Manager software, normalized to the housekeeping gene human RPLP0, and the fold changes in gene expression were calculated relative to the VMs. The detailed step-by-step procedure and primer sequences used can be found in Appendix A.

#### 2.6.5. RNA Sequencing

At day 14 of differentiation, the AMs were either sorted by NKX2.5^eGFP/+^-COUP-TFII^mCherry/+^ selection or lactate purified. The AMs were kept until day 28. The total RNA from 10^6^ AMs per sample was extracted using a NucleoSpin RNA kit (Macherey-Nagel, Düren, Germany) according to manufacturer’s instructions. Libraries and sorted samples were generated from 200 ng of RNA and processed as described in [14]. Differential gene expression (Wald test) between the sorted and lactate purified samples was performed with the R package DESeq2 (v1.34.0) [15]. The genes with an absolute log2 fold change >1.0 and false discovery rate (FDR) *p* < 0.05 were considered differentially expressed and visualized with pheatmap (v1.0.12) and EnhancedVolcano (v1.12.0). Log-scaled normalized counts were used to compare the gene expression levels between samples.

### 2.7. Statistical Analysis

The results were described as means ± S.E.M. An unpaired *t*-test or ordinary one-way ANOVA was applied for the differences in means between conditions, where *p* < 0.05 was considered statistically significant. Detailed statistics are indicated in the figure legend.

See Appendix A for a detailed description.

## 3. Results

### 3.1. An Optimized Protocol to Direct Differentiation of hPSCs toward the Atrial Phenotype by Activation of Retinoic Acid Receptor (RAR)α

During in vivo cardiogenesis, RA is not only crucial for establishing an anteroposterior polarity in the developing heart but also for atrial specification in the lateral plate mesoderm [16,17]. In corroboration, we and others have shown that RA signaling plays a crucial role in the generation of hPSC-derived AMs in vitro. Since atrial specification in vivo occurs following the formation of mesodermal progenitors, we used all-trans RA (atRA) on day 4 of our directed differentiation protocol, which was immediately after the formation of cardiac mesoderm on day 3 but before the expected peak of cardiac progenitors (indicated by activation of cardiac transcription factors such as NKX2.5) at day 7 of differentiation. In the next step, we optimized the concentration of atRA within this time window between day 4 and 7. While in the presence of low concentrations of atRA (10–500 nM), cardiac differentiation from the hPSCs resulted in higher percentages of CMs (10 nM: 9.0 ± 0.2% AMs + 60.3 ± 2.1% VMs; 100 nM: 12.1 ± 1.1% AMs + 54.0 ± 2.0% VMs; 0.5 µM: 20.0 ± 0.5% AMs + 50.7 ± 0.2% VMs), and a higher concentration of atRA (1 µM) enhanced the amount of AMs while reducing the percentage of VMs (45.4 ± 2.7% AMs + 6.4 ± 1.2% VMs) (Figure 1c and Appendix A). Treatment with 3 µM of atRA resulted in a lower percentage of AMs when compared with the 1 µM treatment (36.6 ± 4.5% AMs + 21.0 ± 10.5% VMs) (Figure 1c and Appendix A). However, 10 µM of atRA only led to a small increase in the percentage of AMs and simultaneously also enhanced the amount of VMs in the differentiation (10 µM: 52.9 ± 1.0% AMs + 23.3 ± 1.0% VMs). Only treatment with 20 µM of atRA increased the percentage of AMs, being about 10% without enhancing the number of ventricular counterparts (20 µM: 57.7 ± 0.8% AMs + 8.5 ± 2.0% VMs). Shortening the exposure time of atRA from 72 h to 48 or 24 h had no beneficial effect on atrial differentiation (Figure 1c and Appendix A). Together, this suggested that atRA treatment from day 4 to 7 is optimal to drive atrial specification of mesodermal progenitors during in vitro differentiation. Treatment with either 1 or 20 µM atRA resulted in similar percentages of AMs and VMs. Nevertheless, the percentage of AMs was relatively low, and pure fractions could only be obtained by FACS. RA acts as a ligand for nuclear RA receptors (RARα and RARβ), which in the end activate or repress the transcription of key developmental genes. Thus, to further improve the efficiency of differentiation, we tested the specific induction of RARα or RARβ. Here, we show that the administration of the RARα selective agonist BMS-753 improved the atrial differentiation of hPSCs (Figure 1c, Appendix A). At day 4 of the spin EB differentiation protocol, BMS-753 was used instead of atRA. Similar to the in vivo situation, the exposure time of the selective agonist plays an important role in the induction of atrial specification. We optimized that 20 µM of BMS-753 (but not 1, 10, or 15 µM BMS-753) for only 24 h from day 4 to 5 (but not 48 h) promoted hPSC-derived AMs at a higher efficiency than atRA in hESCs and hiPSCs (Figure 1c and Appendix A), resulting in an average differentiation efficiency of 72% AMs + 5% VMs out of the total cell population (Figure 2e–h). In contrast, at day 14, atRA treatment only resulted in an average purity of 45% AMs + 6% VMs out of the total cell population. When comparing the highest differentiation efficiencies between the atRA and RARα treatments, we showed that in the RARα protocol, 94% of the total cardiomyocyte percentage (77%) was AMs and 6% was VMs, whereas the atRA protocol yielded 88% AMs and 12% VMs for the total cardiomyocyte percentage (51%) (Figure 2e–h and previously reported in [8]). For laboratories that prefer the maintenance of hPSCs on MEF supporter cells, we included a detailed protocol in the supplementary data (Appendix A). Overall, using the RARα protocol, the efficiency of atrial differentiation was significantly increased compared with the atRA protocol (Figure 1c and Figure 2) and also reduced the week-to-week variability of AM subtype differentiation. Treatment with the RARβ selective agonist CD-2314 also promoted comparable percentages of AMs to atRA; however, in our hands, higher variability among multiple differentiations was observed, hindering its use for further applications.

Successful differentiation can be monitored by morphological observation at different steps during differentiation. Crucial steps that may lead to a successful differentiation are the aggregation of EBs one day after EB formation, compact appearance of atRA-treated EBs one day after atRA treatment, and the start of contraction around day 10 of differentiation, with a faster contraction pattern (~90 beats/min) compared with ventricular EBs (~60 beats/min) (Suppl-movie-1_COUP_BMS). The expected differentiation efficiencies for EBs from the atRA protocol were around 40% AMs + 5% VMs and for EBs from the RARα protocol, or around 70% AMs + 5% VMs out of the total cell population. Considering only the cardiomyocyte population, 89% AM + 11% VMs were expected from the atRA protocol and 94% AMs + 6% VMs from the RARα protocol.

If problems during differentiation occur, a list of possible causes and solutions are presented in Table 1.

### 3.2. Enrichment of AMs Based on Metabolic Selection

To obtain highly purified CM populations, several strategies have been described, including a metabolic selection based on media with lactate as a replacement for glucose [18,19,20]. Enrichment of the AMs can be achieved due to their ability to metabolize other energy sources from glucose, as opposed to the glucose-dependent non-cardiomyocyte cell types present in the same culture. With this lactate purification step of 4 days, highly pure AMs (>90% CMs) can be obtained consistently and kept in a cardiac maturation medium as needed for downstream applications [21] (Figure 3). The purification was less efficient in EB format, most likely because of the limited diffusion of the medium to the inner core of the EBs. Moreover, we experienced that when the EBs were maintained for longer than 21 days in EB format and exposed to the cardiomyocyte medium, their dissociation became more difficult, resulting in higher cell loss. Therefore, we included an intermediate step of single-cell seeding so the medium was equally accessible to all cells and the recovery of CMs was easier and less stressful for the cells.

The expected yield was, on average, 23,000 ± 6083 cells from one EB with 5000 hPSCs (4.6 ± 1.2 cells per hPSC), of which 10,350 ± 2737 (2.07 ± 0.55 AMs per hPSC) or 16,120 ± 4318 (3.22 ± 0.86 AMs per hPSC) were AMs if the atRA or RARα protocol was followed, respectively (Figure 3c).

After lactate purification, more than 90% of the total cell population was expected to be AMs in RARα-induced EBs. Approximately 25% cell loss from the total population was expected. In terms of absolute quantification, this resulted in 17,250 ± 7902 cells per seeded EB of 5000 hPSCs (3.45 ± 1.58 cells per hPSC), of which 15,542 ± 4121 were AMs (3.11 ± 0.82 AMs per hPSC), indicating that only 3.6% of the AMs were lost during the dissociation or metabolic purification step (Figure 3c).

### 3.3. Characterization of AMs

The dual COUP-red (NKX2.5^eGFP/+^-COUP-TFII^mCherry/+^) fluorescent hPSC reporter allowed characterization of AMs by flow cytometry analysis of the endogenous reporters (Figure 1 and Figure 2). We confirmed that sorted NKX2.5-GFP (G^+^) and COUP-TFII-mCherry-positive (M^+^) AMs from the COUP-red line not only expressed sarcomere protein cardiac Troponin T but also stained positive for NKX2.5 and COUP-TFII, and the majority of the AMs expressed both transcription factors (Figure 4a). Thus, when using other hPSC lines, CMs can be characterized by immunotyping or flow cytometry for cardiac sarcomere markers, such as cardiac troponin I or T (Figure 4a and Appendix A). For evaluation of the atrial phenotype, immunofluorescent staining for atrial-enriched COUP-TF I or II could be performed (Figure 4a,b, Appendix A). A combination of both markers—cardiac troponin with COUP-TF I or II—enabled characterization in the hPSC lines without the use of an atrial or ventricular reporter.

At a molecular level, the genes preferentially expressed in AMs were comparable in the RARα- or atRA-induced AMs (Figure 5a). Interestingly, *PITX2*, a transcription factor that is expressed in the left atrium [22], was expressed significantly more in the RARα-induced AMs. Similarly, the ventricular markers were downregulated in both the RARα- and RA-induced AMs compared with the VMs. These results indicate that treatment with RARα yields CMs with an atrial identity and that the molecular profile is comparable to that of atRA-AMs. Next, we performed transcriptional analysis by RNA sequencing on sorted and lactate-purified RARα-induced AMs in order to verify that the metabolic purification did not lead to different atrial identities based on the gene expression levels. First, as shown by a sample-to-sample distance heatmap (Figure 5b), the overall gene expression between the sorted and lactate-purified samples showed a close relationship, as observed by similar clustering of the samples from each treatment, suggesting that the transcriptome profiles from the sorted and lactate-purified AMs were similar. Following differential expression analysis revealed 321 differentially expressed genes (DEGs), of which 85 were upregulated and 236 were downregulated in the sorted AM (log2 fold change > 1 and false discovery rate (FRD)-corrected *p*-value < 0.05; Appendix A). The relevant upregulated genes in the sorted AMs included *ACTA1* and potassium channels *KCNT2* and *KCNH7* (Figure 5c). Overall, the genes predominantly expressed in AMs or VMs and encoding for sarcomeric proteins, ion channels, and transcription factors and involved in energy metabolism were equally expressed in the AMs from both treatments (Figure 5d–h). Specifically, the expression levels of atrial and ventricular genes such as *MYL7*/*MYL2* (15.70 counts in sorted and 15.65 in lactate vs. 8.82 and 9.85, respectively) and *MYH6*/*MYH7* (18.14 and 18.32 vs. 13.67 and 13.04, respectively), together with high expression of atrial-specific markers such as *NPPA* (15.18 counts in sorted and 14.46 in lactate), *COUP-TFII* (11.4 and 11.2) and potassium channels *KCNA5* (10.88 and 10.93), *KCNJ3* (9,92 and 9,53), and *KCNJ5* (9.01 and 8.21), confirmed that the AMs enriched by lactate purification maintained their atrial identities. Of note, genes encoding for sarcomeric proteins such as *MYOM2* (5.35 counts in sorted vs. 7.05 in lactate), *MYOM3* (5.66 vs. 8.48), and *TNNT3* (4.18 vs. 7.52) were significantly higher in the lactate-purified AMs (Figure 5c,e), suggesting that contractile function was maintained or even improved (possibly more mature). Similarly, both major connexin isoforms of the working myocardium encoded by the genes *GJA1* (connexin 43) and *GJA5* (connexin 40) were higher expressed in lactate-purified AMs compared with the sorted AMs (GJA5: 5.66 counts in sorted vs. 7.81 in lactate; GJA1: 9.58 vs. 10.74). Importantly, the increase in the atrial isoform *GJA5* was more pronounced than that of the ventricular isoform *GJA1*. Finally, the expression of *CKM*, which encodes for a cytoplasmic creatine kinase that plays a role in energy transduction, and *CPT1A,* which encodes for the enzyme carnitine-O-palmitoyl transferase I involved in fatty acid oxidation, was also higher in the lactate AMs, suggesting a difference in energy use between both conditions (Figure 5h). Together, these findings confirmed that the RARα AMs had an atrial identity and that the AMs maintained their atrial phenotypes and functionalities after metabolic purification utilizing lactate.

Moreover, electrophysiological characterization could be performed to further confirm the atrial phenotype and evaluate the functionality of the AMs (Figure 6). As already described by patch clamping [8], the AMs showed a faster repolarization compared with the VMs (Figure 6a,b) and higher beating frequencies when quantified optically using the voltage-sensitive dye FluoVolt (Figure 6). We did not observe significant differences in the action potential duration (APD) (Figure 6a,b) (APD20 RARα: 186.7 ± 10.3 ms; APD20 RA: 205.4 ± 12.6 ms; APD20 VMs: 255.2 ± 15.3 ms; APD50 RARα: 247.1 ± 15.3 ms; APD50 RA: 253.7 ± 18.9 ms; APD50 VMs: 422.1 ± 14.8 ms; APD90 RARα: 389.2 ± 24/4 ms; APD90 RA: 371.3 ± 13.7 ms; APD90 VMs: 733.9 ± 28.2 ms;) or contraction frequency in AMs from the atRA or RARα protocol (Figure 6c,d) (RARα: 1.5 ± 0.1 Hz; RA: 1.6 ± 0.1 Hz; VMs: 0.7 ± 0.0 Hz).

## 4. Discussion

Efficient differentiation procedures for human CMs from hPSCs have great potential for a wide range of applications, from disease modeling to regenerative medicine. For this purpose, efficient and robust procedures to differentiate hPSCs toward specific cardiac subtypes such as VMs or AMs are needed, which can then be used as pure monotypic cultures or in combination with other cell types in controlled ratios for downstream applications. In our previous work, we showed that the use of hPSC-derived AMs can effectively predict the atrial selectivity of pharmacological compounds and thus can successfully be applied to pre-clinical screening of anti-arrhythmic agents. A patch clamp of single AMs showed evidence that AMs—but not VMs—respond to pharmacological manipulation of atrial-selective potassium I_Kur_ and I_KAch_ currents [7]. In addition, we have also shown that differentiation of hPSC-derived AMs could serve as a valid model for investigation of the developmental aspects of human atrial specification in vitro by knockout of COUP-TFII. Via CRISPR/Cas9-mediated knockout, we showed that COUP-TFII is not required for early atrial specification in hPSCs [8]. Moreover, several groups have shown that hPSC-derived AMs are valuable for modeling atrial fibrillation by developing AM cell patches or rings [23,24,25]. Optical imaging of the atrial tissues showed the development of re-entrant rotor patterns and allowed screening of anti-arrhythmic agents. Recently, Lemme et al. generated engineered heart tissues from AMs, which allowed electrophysiological, contractile, and morphological analysis of 3D atrial cardiac tissues in healthy and disease conditions to mimic patient-specific conditions [26].

Differentiation to AMs and VMs yielded heterogeneous mixtures of CMs, in which the AMs were accompanied by a smaller percentage of VMs and vice versa. Since specific cell surface markers for purification of atrial and ventricular CMs are lacking, it is pivotal to develop efficient and standardized differentiation protocols. Use of cardiac-subtype specific genetic reporter lines allows identification and selection of AMs and VMs. In 2014, a bacterial artificial chromosome reporter construct with fluorescent expression under control of the atrial-specific gene sarcolipin was developed and used for the identification of hPSC-derived AMs [27]. Recently, we incorporated a targeting construct including sequences encoding for the red fluorescent protein mCherry at the COUP-TFII genomic locus in human embryonic stem cells (hESCs) expressing GFP from the NKX2.5 locus to select pure populations of atrial and ventricular CMs [8]. This dual NKX2.5^eGFP/+^-COUP-TFII^mCherry/+^ (COUP-red) reporter allows for the identification and selection of GFP^+^/mCherry^+^ AMs, as well as GFP^+^/mCherry^-^ VMs with the transcriptional and functional properties of hPSC-derived AMs or VMs, respectively, from heterogeneous cardiac differentiation cultures. The use of this reporter line allowed further improvement of the differentiation efficiencies by fast live cell fluorescent analysis.

Current protocols for efficient differentiation toward AMs are dependent on the formation of EBs, which limits the large-scale production of AMs and complicates subsequent assays. Nevertheless, AM differentiation via the spin EB method has the advantage of better resembling embryonic development regarding their three-dimensional (3D) shapes over the monolayer-based protocols. Therefore, we sought to describe a detailed spin EB protocol for differentiating either hPSC cultured on feeder layers or in feeder-free cultures toward AMs with metabolic selection and follow-up characterization of the resulting AMs. When optimizing the concentration and time window of atRA treatment, we did not identify a condition that resulted in an increased percentage of AMs after differentiation without simultaneously increasing the percentage of VMs. Thus, to further optimize AM differentiation, we focused on modulating the RA pathway by activating the RA receptors alpha (RARα) or beta (RARβ). Similar to atRA stimulation, the activation of RARβ signaling only had a minor effect on the total yield of AMs in hPSC differentiation. Only the induction of RARα tremendously increased the AM fraction from 45% to more than 72%. Interestingly, only shortening the induction time frame from 72 h to 24 h had a beneficial effect on the AM specifications. This new method yielded 30% more AMs than in the previously described atRA-based differentiation protocols and increased the yield of AMs per hPSC from 2.07 to 3.22, yielding approximately 1 million AMs per plate with 60 EBs. The resulting AMs obtained with RARα activation presented similar gene expression patterns, electrophysiological properties, and immunohistochemistry when compared to atRA-induced AMs. Although the resulting efficiency with RARα was higher than that with atRA, it did not generate pure populations of AMs, which is desired for pre-clinical and clinical applications. Therefore, we also complemented our protocol with a detailed metabolic selection method to further purify and isolate the AMs. This metabolic selection based on lactate instead of glucose in the culture medium further enhanced the AM percentage from 72% to more than 90% without compromising the AM yield (15,542 per EB, equal to 3.11 per hPSC) or presumed functionality of the AMs based on RNAseq analysis.

In conclusion, we presented a robust and detailed spin EB protocol for generating AMs with high efficiency and yield, which will be important for their application in atrial-related studies, from modeling atrial diseases to drug screening of atrial-selective compounds and tissue engineering.

## Figures and Tables

**Figure 1 jpm-12-00628-f001:**
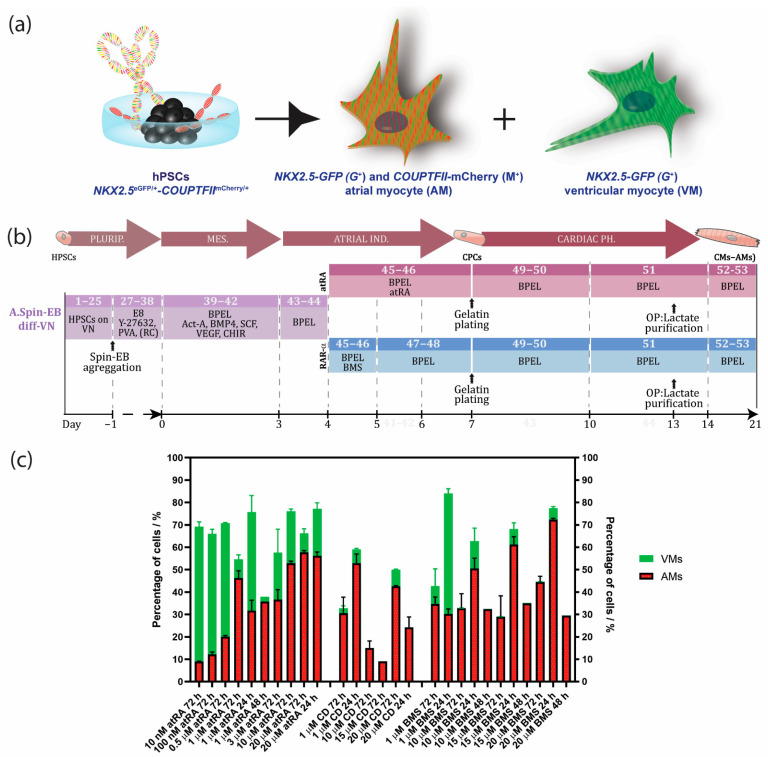
Timeline with overview of hPSC-derived atrial differentiation protocols and NKX2.5^eGFP/+^-COUP-TFII^mCherry/+^ fluorescent reporter hPSC line for selection of atrial (AMs) and ventricular cardiomyocytes (VMs). (**a**) Visualization of the NKX2.5^eGFP/+^-COUP-TFII^mCherry/+^. AMs can be selected by NKX2.5-GFP (G^+^) and COUP-TFII-mCherry (M^+^) expression, whereas VMs can be purified by NKX2.5-GFP (G^+^) expression and the absence of COUP-TFII-mCherry (M^−^). (**b**) Spin EB protocol with RA or RARα induction from hPSCs cultured in defined culture conditions with E8 medium on vitronectin. Numbers in each section indicate the step corresponding to the step-by-step protocol of Appendix A. (**c**) Optimization of the atrial differentiation protocol by activation of all-trans retinoic acid (atRA), retinoic acid receptor alpha (RARα) with BMS-753 (BMS), or RARβ with CD-2314 (CD) using the NKX2.5^eGFP/+^-COUP-TFII^mCherry/+^ line for quantifying the percentage of AMs or VMs. Data shown as means ± S.E.M. Plurip. = pluripotency; MES. = mesoderm; Ind. = induction; Ph. = phenotype; CPCs = cardiac progenitor cells; CMs (AMs) = atrial cardiomyocytes; VN = vitronectin; Diff = differentiation; Act-A = activin-A; RC = RevitaCell; OP = optional.

**Figure 2 jpm-12-00628-f002:**
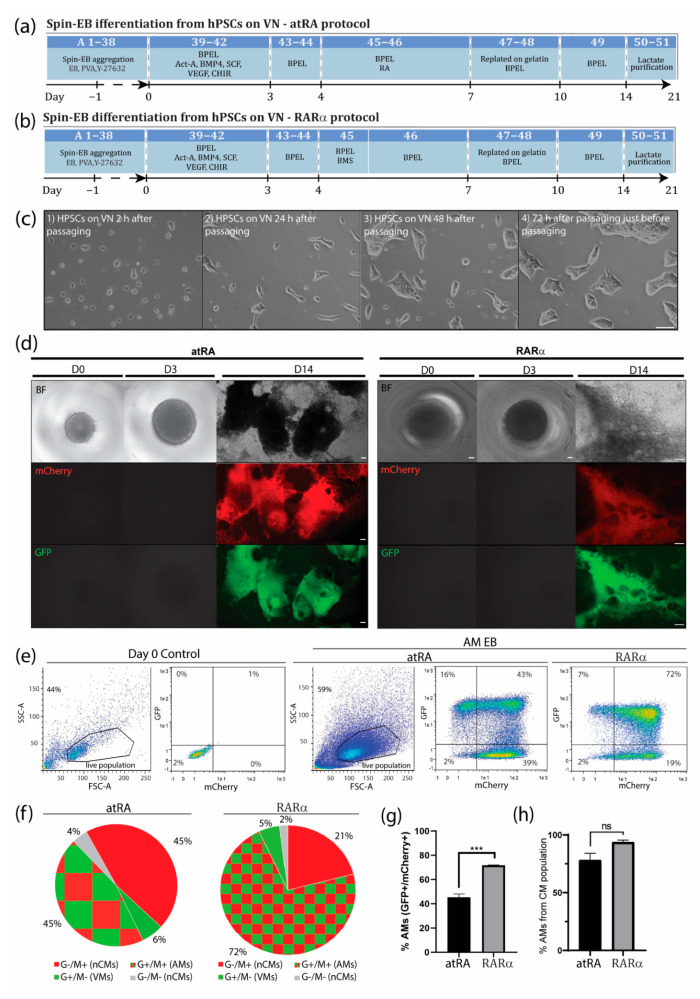
Improved atrial differentiation by activation of retinoic acid receptor (RAR)α. (**a**) Schematic step-by-step representation of atRA-mediated atrial differentiation toward AMs. Numbers correspond to step-by-step procedure in Appendix A. (**b**) Schematic step-by-step representation of RARα-mediated atrial differentiation toward AMs. Numbers correspond to step-by-step procedure in Appendix A. (**c**) (1) HPSCs cultured in E8 medium on VN 3 h after passage and (2) 24 h, (3) 48 h, and (4) 72 h after passage and just before the next passage. Scale bar = 100 µm. (**d**) Spin EBs from hPSCs grown in E8 on VN at days 0, 3, and 14 of differentiation with atRA (left panel) or optimized atrial protocol with RARα (right panel). Endogenous COUP-TFII-mCherry expression together with endogenous NKX2.5-GFP and bright-field (BF) images in AM and VM differentiations. Scale bar = 100 µm. (**e**) Representative flow cytometry plots depicting the percentage of NKX2.5-GFP (G^+^) or COUP-TFII-mCherry-positive (M^+^) cells at day 14 in AM RA- or RARα-induced differentiation in spin EB format. (**f**) Pie chart representation of average flow cytometry percentages of NKX2.5-GFP (G^+^) or COUP-TFII-mCherry-positive (M^+^) cells and their counterparts in atRA or RARα differentiations (n = 3). AMs = atrial cardiomyocytes; VMs = ventricular cardiomyocytes; nCMs = non-cardiomyocytes. (**g**) Average percentage of AMs differentiated with atRA or RARα protocol, indicated as percentage of NKX2.5-GFP and COUP-TFII-mCherry-positive respective total cell population (n = 3, *** *p* < 0.001). (**h**) Average percentage of AMs differentiated with atRA or RARα protocol, indicated as percentage of NKX2.5-GFP and COUP-TFII-mCherry-positive cells respective of the cardiomyocyte population (n = 3, ns = not significant).

**Figure 3 jpm-12-00628-f003:**
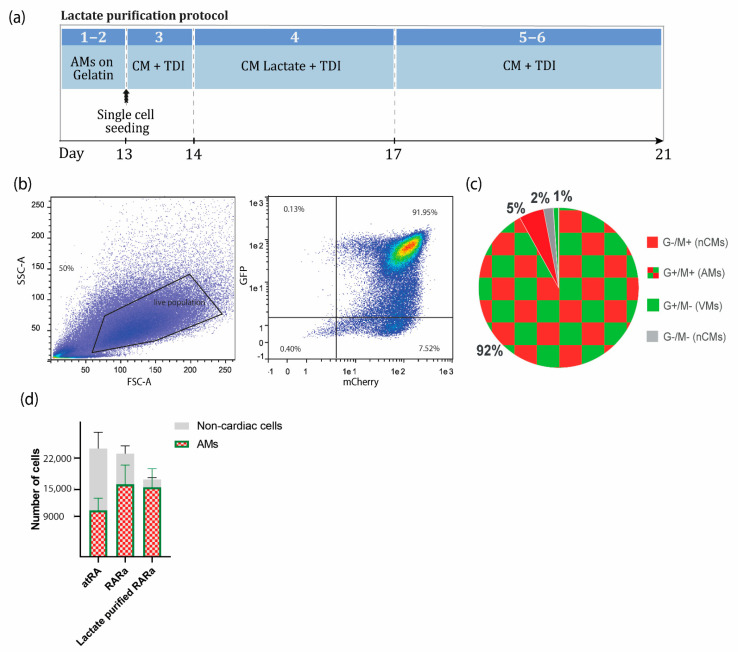
Metabolic selection of AMs. (**a**) Schematic representation of the lactate protocol from COUP-red RARα-induced AMs. Steps correspond to the step-by-step section ‘D. Enrichment of AMs based on metabolic selection’ in Appendix A. (**b**) Representative flow cytometry plots depicting the percentage of NKX2.5-GFP and COUP-TFII-mCherry-positive AMs at day 21 in RARα-induced differentiation in spin EB format, followed by enrichment via lactate purification. (**c**) Pie chart representation of average flow cytometry percentages of NKX2.5-GFP (G^+^) and COUP-TFII-mCherry-positive (M^+^) AMs, G^+^/M^−^ VMs, and G^−^/M^−^ or G^−^/M^−^ non-cardiac cells (n = 3). (**d**) Yield of atrial differentiation in atRA, RARα, and lactate-purified RARα. Data plotted as means ± S.E.M. (n = 3). AMs = atrial cardiomyocytes; VMs = ventricular cardiomyocytes; nCMs = non-cardiomyocytes; CM + TDI = cardiomyocyte medium plus TDI; CM Lactate + TDI = cardiomyocyte medium for lactate purification plus TDI.

**Figure 4 jpm-12-00628-f004:**
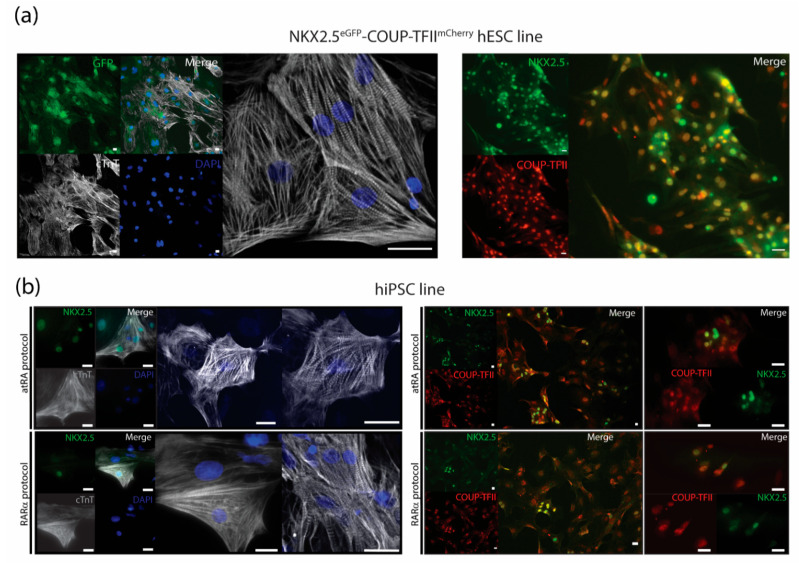
Immunostainings for characterization of AMs. (**a**) Cardiac troponin T (cTnT) together with endogenous NKX2.5-GFP and nuclear stain DAPI with representative pictures of immunostainings for COUP-TFII together with NKX2.5 in RARα-treated AMs at day 21 in unsorted cultures after dissociation and re-plating. Scale bar = 25 µm. (**b**) Representative pictures of immunostainings for cardiac troponin T (cTnT) together with NKX2.5 and nuclear stain DAPI and immunostainings of COUP-TFII together with NKX2.5 of atRA and RARα-treated, hiPSC-derived AMs in sorted cultures at day 21. Scale bar = 25 µm.

**Figure 5 jpm-12-00628-f005:**
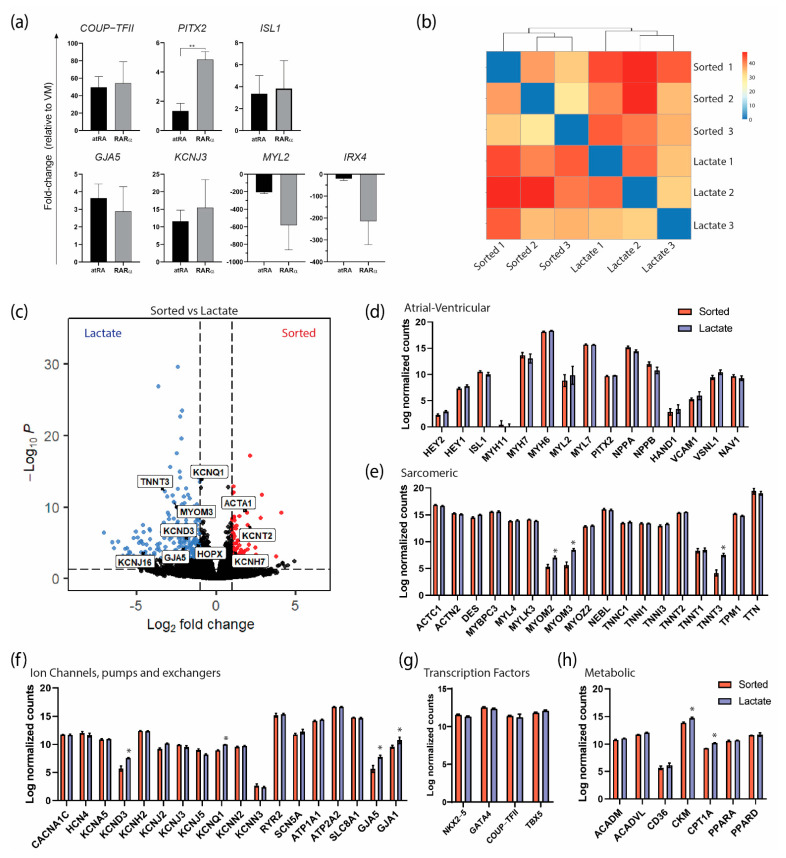
Transcriptional characterization of atRA and RARα COUP−Red AMs by RNAseq and RT−qPCR. (**a**) Transcriptional profiling of selected atrial-specific genes by qPCR in G^+^/M^+^ cells from atRA or RARα protocol relative to expression in ventricular cardiomyocytes (VM). ** *p* < 0.01. Data shown as means ± S.E.M. Significance determined with unpaired *t*-test (n = 3 biological replicates per condition). (**b**) Expression heatmap of sample-to-sample distances on the matrix of variance-stabilized data for overall gene expression (n = 3 biological replicates per condition). (**c**) Volcano plot of gene expression in sorted vs. lactate AMs. Genes of interest are labeled. (**d**−**h**) Expression of genes (log-transformed normalized counts) predominant in atrial or ventricular cardiomyocytes (**d**), encoding for cardiac sarcomeric proteins (**e**), ion channels, pumps and exchangers (**f**)**,** and transcription factors (**g**) and involved in energy metabolism (**h**). Means shown with error bars indicating S.E.M. * *p* < 0.05, two-way ANOVA.

**Figure 6 jpm-12-00628-f006:**
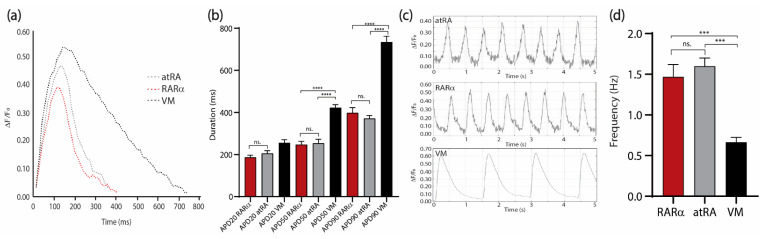
Electrophysiological characterization of sorted COUP-red atRA and RARα-induced AMs compared to sorted VMs at day 21 of differentiation. (**a**) Changes in AP morphology indicated by changes in fluorescence intensity of the FluoVolt AP indicator over time in atRA (dotted gray) and RARα (dotted red) AMs and VMs (dotted black). (**b**) Comparison of mean AP duration at 20% (APD20), 50% (APD50), and 90% (APD90) repolarization in atRA and RARα AMs and VMs (n = 6 per condition). (**c**) AP waveforms indicated by changes in fluorescence intensity of the FluoVolt AP indicator over time in atRA (upper panel) and RARα (middle panel) AMs and VMs (lower panel). (**d**) Contraction frequency of atRA or RARα AMs and VMs obtained from AP waveforms (n = 6 cells); ns. = not significant. *** *p* < 0.001. **** *p* < 0.0001. All values are represented as means ± S.E.M. One-way ANOVA test is shown. AMs = atrial cardiomyocytes; VMs = ventricular cardiomyocytes.

**Table 1 jpm-12-00628-t001:** Troubleshooting of different problems that may occur during differentiation steps.

Step	Problem	Possible Cause	Possible Solution
EBs do not form	PVA solution	PVA solution too old or frozen	Prepare fresh PVA solution
EBs do not attach	(1) Gelatin solution (2) Wells plate brand	(1) Old gelatin solution (2) Adherance of gelatin to the well	(1) Prepare fresh 0.1% gelatin solution (2) Change to indicated wells plate brand or plasma-treat the plate and coat afterwards
EBs do not beat	Differentiation into CMs did not work efficiently	(1) Concentration of growth factors (2) HPSC passage number too high	(1) Perform titration of growth factors (2) Start lower passage of hPSCs
CMs have no atrial identity	RA or BMS concentration	Old RA or BMS solution	Prepare fresh RA or BMS aliquots or order new RA or BMS
CMs die during dissociation	CMs cannot handle the dissociation	(1) Long incubation time with TrypLE (2) Pipetting too harsh	(1) Shorter incubation period (2) Pipetting more carefully

## Data Availability

The RNA-sequencing data discussed in this publication are available at the NCBI’s Gene Expression Omibus (GEO) under GEO accession number GSE178473 and GSE200662.

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
