# Peer review of "Improved Atrial Differentiation of Human Pluripotent Stem Cells by Activation of Retinoic Acid Receptor Alpha (RARα)"

_jpm, 2022, doi:10.3390/jpm12040628_

Round 1
Reviewer 1 Report
The authors showed the protocol of differentiating human iPSCs to atrial myocytes. The selection of and/or differentiation into specific subtype of cardiomyocytes has been demanded in this research field. From this point, the authors work may have a certain impact in this field. The authors further characterized their differentiated AMs.
This reviewer thinks that this manuscript is publishable.
I have a few comments before publication.
Line 177 atRA → all trans RA (atRA)
This reviewer thinks that the authors had better expand discussion the expression of COUP-TFⅡand NKX2.5, and further genes that are involved in the development of AMs so that the readers can see how these TFs play roles in AM development and may have an idea in what differentiation stage these TFs work.
Reviewer 2 Report
The manuscript of Schwach et al. describes the optimization of a directed differentiation of hPSCs towards cardiomyocytes with atrial identity by modulating the retinoic acid signaling cascade. Authors claim that this protocol improves upon past protocols by selectively inhibiting RARα using the selective agonist BMS-753.
Overall, the manuscript reads as a protocol tentatively adapted as an experimental article. Likely because of this, the flow of the manuscript is somewhat uncanny. In addition, the “optimization” performed claimed by the authors is insufficiently justified by the results shown.
Major Issues:
- Abstract is vague and does not highlight key findings. (e.g. % of differentiation obtained and /or gained by the optimization; use of metabolic selection etc)
- In the majority of figures and in text authors often talk about GFP and mCherry. This can bring a lot of confusion. Assuming that GFP positive always means NKX2.5 positive and mCherry positive always means COUP-TFII positive can facilitate reading. Nevertheless, sometimes the manuscript creates some doubts if these are in fact the markers present or if there are other sources of GFP/mCherry. Authors should include in the text and figures the gene corresponding to the GFP and mCherry throughout the manuscript. This can significantly help a reader that is not familiarized with the original work.
- The claim of a process optimization dissipates since the concentration range or activations at different days are not showed or properly compared. Authors claim that low concentrations of RA (1–10 nM) resulted in higher CM percentage but 1 μM enhanced the expression of atrial markers. The results for the 1–10 nM are not shown. In addition, from 10 nM to 1 μM there is a long margin of optimization. And why not more? 1 μM is also used by other protocols (e.g Pei et al. 2017 that authors cite) thus I fail to see any type of optimization in the results showed.
- Similar to point 3, authors claim that they optimized the use of the RARα agonist BMS-753 to promote a higher differentiation efficiency. There are several problems with this. (1) It is hard to understand why authors do not show what happens with the RARβ agonist. (2) Once again, as in point 3, optimization of the concentrations is not shown or justified. (3) Comparison with the atRA is unfair, since concentrations of the BMS-753 are 20x higher and activation timings are not equal. What happens if atRA is used for only 24h at a higher concentration?
- Authors never mention the amount of CMs obtained and/or the yield per hiPSCs. Percentages can be meaningless if the final result is a significant loss of yield.
- Figure 2 and a part of the paper describe the differentiation of AMs from hPSCs cultured on MEFs. Authors do not quantify this protocol and/or compare this protocol to the one without MEFs. In addition, there is a one-day culture in Matrigel prior to EB formation to start the differentiation. Authors do not appear to justify in any way the importance of this in the context of an experimental article (maybe a variation or adaptation of the protocol for labs that culture hPSCs in MEFs?).
- Regarding the results of metabolic purification (figure 7). What is the rational for single-cell seeding prior to this step? How many independent experiments were performed? How many cells are lost during the 3-day exposure to lactate? Once again percentage of positive cells is meaningless if at the end of the procedure non-functional or a low number of functional cells are obtained.
Minor Issues:
- Email of one of the corresponding authors appears incorrect.
- The term “Quantitative real-time PCR (RT-qPCR)” is starting to be used more in the literature but authors still need to emphasize that what they do is relative expression (as it appears) and not absolute quantification.
- Figure subpanels sometimes are not called out in the text in the order of appearance. e.g. figure 2D and 2E are only referenced after figure 3 and figure 4.
- How were the VMS sorted? (Figure 6) GFP+ and mCherry negative?
Round 2
Reviewer 2 Report
Authors performed a remarkable job in addressing all points (some of them I believed were challenging) with additional results, new data presentation and interpretation, and additional justifications. Presentation and flow were improved. Thus, I have no further comments/issues.
Finally, I would like to congratulate the authors for the effort and perseverance in the revision of the manuscript and wish them the best for future endeavors.